# Intestinal Helminths in Wild Rodents from Native Forest and Exotic Pine Plantations (*Pinus radiata*) in Central Chile

**DOI:** 10.3390/ani11020384

**Published:** 2021-02-03

**Authors:** Maira Riquelme, Rodrigo Salgado, Javier A. Simonetti, Carlos Landaeta-Aqueveque, Fernando Fredes, André V. Rubio

**Affiliations:** 1Departamento de Ciencias Biológicas Animales, Facultad de Ciencias Veterinarias y Pecuarias, Universidad de Chile, Santa Rosa 11735, La Pintana, Santiago 8820808, Chile; mairariquelme@gmail.com (M.R.); rodrigo.salgadomoya@gmail.com (R.S.); 2Departamento de Ciencias Ecológicas, Facultad de Ciencias, Universidad de Chile, Casilla 653, Santiago 7750000, Chile; jsimonet@uchile.cl; 3Facultad de Ciencias Veterinarias, Universidad de Concepción, Casilla 537, Chillán 3812120, Chile; clandaeta@udec.cl; 4Departamento de Medicina Preventiva Animal, Facultad de Ciencias Veterinarias y Pecuarias, Universidad de Chile, Santa Rosa 11735, La Pintana, Santiago 8820808, Chile; ffredes@uchile.cl

**Keywords:** Chile, helminths, land conversion, Rodentia, small mammals, zoonosis, wildlife

## Abstract

**Simple Summary:**

Land-use changes are one of the most important drivers of zoonotic disease risk in humans, including helminths of wildlife origin. In this paper, we investigated the presence and prevalence of intestinal helminths in wild rodents, comparing this parasitism between a native forest and exotic Monterey pine plantations (adult and young plantations) in central Chile. By analyzing 1091 fecal samples of a variety of rodent species sampled over two years, we recorded several helminth families and genera, some of them potentially zoonotic. We did not find differences in the prevalence of helminths between habitat types, but other factors (rodent species and season of the year) were relevant to explain changes in helminth prevalence. Given that Monterey pine plantations are one of the most important forestry plantations worldwide, and due to the detection of potentially zoonotic helminths, more research should be conducted in this study area and elsewhere in order to better understand the effect of pine plantations on parasites and pathogens in rodents and other wildlife hosts.

**Abstract:**

Native forests have been replaced by forestry plantations worldwide, impacting biodiversity. However, the effect of this anthropogenic land-use change on parasitism is poorly understood. One of the most important land-use change in Chile is the replacement of native forests by Monterey pine (*Pinus radiata*) plantations. In this study, we analyzed the parasitism (presence and prevalence) of intestinal helminths from fecal samples of wild rodents in three habitat types: native forests and adult and young pine plantations in central Chile. Small mammals were sampled seasonally for two years, and a total of 1091 fecal samples from seven small mammal species were analyzed using coprological analysis. We found several helminth families and genera, some of them potentially zoonotic. In addition, new rodent–parasite associations were reported for the first time. The overall helminth prevalence was 16.95%, and an effect of habitat type on prevalence was not observed. Other factors were more relevant for prevalence such rodent species for *Hymenolepis* sp. and season for *Physaloptera* sp. Our findings indicate that pine plantations do not increase helminth prevalence in rodents compared to native forests.

## 1. Introduction

Anthropogenic land-use change can impact biodiversity and human health, being a major driver of biodiversity loss and zoonotic disease emergence [1] Land-use change can modify host–parasite interactions through a variety of mechanisms that involve changes in abundance, behavior and immune response of hosts, as well as the composition and structure of host community [2]. Additionally, land-use change can modify abiotic conditions, which, in turn, may influence the transmission and life cycle of parasites such as helminths, which have several life stages with close contact with the environment [3]. 

Rodents are one of the most important reservoir hosts of zoonotic pathogens [4], and reservoir species are commonly found in high abundance in anthropogenic-modified habitats (e.g., agricultural lands, pasture lands) [5]. Wild rodents can act as definitive, intermediate and paratenic hosts of several endoparasites (helminths and protozoa) with zoonotic potential, such as *Capillaria hepatica* Bancroft 1893, *Cryptosporidium* spp., *Giardia* spp., *Toxoplasma gondii* Nicolle and Manceaux 1908, *Schistosoma* spp., etc. [6,7]. In addition, helminths are the most prevalent group of macroparasites in rodents [8]. Therefore, the study of the effect of anthropogenic land-use change on gastrointestinal helminths in rodents is needed to better understand host–parasite interactions in wildlife with public health implications.

The forest industry constitutes an important economic activity in tropical and temperate regions of developing countries, replacing large areas of native forests and grasslands by plantations of fast-growing exotic trees [9]. Despite the relevance of plantations worldwide, the study of the effect of monoculture plantations on wildlife parasites has been scarce [2], but some studies have found that plantations may increase the abundance of some ectoparasitic mites in rodents [10] as well as parasite richness in several wild mammals [11]. An important plantation in the global forest industry is Monterey pine, *Pinus radiata* (D. Don, 1836), accounting for 32% of productive plantations worldwide [12]. In Chile, Monterey pine plantations are one of the most important land uses in south-central regions [13], covering approximately 1.9 million ha and accounting for 68% of forestry plantations of the country (http://www.corma.cl). This forest plantation modifies biodiversity, including the composition and structure of rodents [14,15,16], as well as the prevalence and load of mites, *Ornithonyssus* sp. (Mesostigmata), in rodent hosts [10]. 

Several gastrointestinal helminths have been detected in native and exotic rodents in Chile [17,18,19,20,21]. However, to our knowledge, the effect of land use on intestinal helminths of rodents inhabiting monoculture plantations has not been investigated. Therefore, the aim of this study was to describe the presence and compare the prevalence of gastrointestinal helminths in feces from wild rodents inhabiting forestry plantations and native forests in a highly disturbed landscape from central Chile. 

## 2. Materials and Methods 

### 2.1. Ethical Statement

All procedures for trapping and handling rodents followed the guidelines of the American Society of Mammalogists for the use of wild mammals in research [22] and followed standardized safety guidelines recommended by the Centre for Disease Control and Prevention [23]. Sampling procedures were authorized by the Servicio Agrícola y Ganadero (SAG; Chilean Fish and Wildlife Service) (License No. 6831/2015) and the Ethics Committee of the Faculty of Science, University of Chile.

### 2.2. Study Area and Sampling Sites

The study was conducted in Tregualemu (35°58′ S, 072°44′ W), located in the coastal range of the Maule Region in central Chile. The landscape contains interspersed remnants of native forest and extensive stands of Monterey pine of different ages [24]. The native forest of the study area includes the Queules National Reserve and contiguous forests that make up an area of 600 ha. The forest is composed of *Nothofagus glauca* (R. Phil.) Krasser 1896 and *N. obliqua* (Mirb.) Oerst, 1871, as dominant species and the evergreen *Cryptocarya alba* (Molina) Looser 1950 and *Peumus boldus* Molina, 1782. Monterey pine plantations in this area are managed by maintaining a developed understory or multiple vegetation strata within monocultures [25]. Understory vegetation in mature pine plantation contains *Aristotelia chilensis* Stunz 1914, *P. boldus* with exotic species such as *Genista monspessulana* (L.) L.A.S.Johnson 1962 and *Rubus ulmifolius* Schott, 1818 [26].

Small mammals were sampled in three dominant habitat types: (1) native forest, (2) adult pine plantations (>15 years old) and (3) young pine plantations (3–4 years old). A total of 12 sampling sites were selected: three sites each in native forest and adult pine plantations and six in young pine plantations. Each site was separated by at least 400 m from each other (mean distance between sites = 2025 m). 

### 2.3. Small Mammals and Fecal Sampling

Small mammals were sampled once each season (summer: January; autumn: April–May; winter: July–August; spring: November) over 2 years (2016–2017). Each site was sampled in all periods, except for one adult pine plantation and one young pine plantation site that were not sampled in autumn 2017 due to logistical constraints. At each site, live Sherman traps were placed separated by 10 m, forming a 7 × 10 grid (70 traps), for four consecutive nights. All traps were baited with rolled oats and vanilla essence and checked daily at dawn. After capture, animals were identified to species [27], measured, weighed, sexed and marked with uniquely numbered ear tags (National Band and Tag Co., Newport, KY). Fecal samples were collected directly from the animal or from the trap and then preserved in 70% ethanol in 2-mL microtubes. After handling, animals were released in the same place they were captured. 

### 2.4. Parasitological Analysis

The presence of helminth eggs and larvae in feces was analyzed using the routine coprological method of modified Telemann [28,29]. Briefly, each stool sample was placed in a 15-mL Falcon tube with 10 mL of 70% ethanol and 2 mL of diethyl ether and centrifuged at 2000 rpm for 10 min at room temperature. After centrifugation, the supernatant was removed and 100 μL of the sample was placed on a glass microscope slide, and then, a 24 × 24-mm cover slip was placed on the surface of the sample. Finally, slides were scanned under 10× and 40× objective lenses of a light microscope. Identification of helminths was based on published taxonomic keys [28] and using references of eggs collected from adult helminths by Landaeta-Aqueveque et al. [30]. Laboratory analyses were performed at the Parasitology Laboratory, Faculty of Veterinary Science, University of Chile, and at the Parasitology Laboratory, Faculty of Veterinary Science, University of Concepcion.

### 2.5. Data Analysis

Prevalence (the proportion of positive animals divided by the total number tested) was calculated and 95% confidence intervals (CI) were determined using the Clopper–Pearson method [31] for each rodent species. These analyses were performed using the software Quantitative Parasitology v.3.0 [31]. Generalized linear models (GLMs) with binomial distribution and logit function were used to identify variables that may explain prevalence. Models were built using all parasites combined and also using the most frequent helminths separately. The explanatory variables analyzed were habitat type, season and rodent species. For each multivariate model, we calculated Akaike information criterion adjusted for sample size (AICc), differences in AICc and Akaike weights (wr). The best models were based on the lowest AICc values [32]. Selected models were validated by the Hosmer–Lemeshow goodness-of-fit test. We performed these analyses with R software (R Core Team 2017), including packages “MuMIn” [33] and “ResourceSelection” [34].

## 3. Results

### 3.1. Small Mammal Sampling

A total of 1962 individuals were captured. Small mammals belonged to seven rodent species and a marsupial, the elegant fat-tailed mouse opossum *Thylamys elegans* (Waterhouse, 1839). The most common species captured were *Abrothrix olivacea* Waterhouse, 1837 (56.5%), *Oligoryzomys longicaudatus* Bennett, 1832 (20.3%) and *Abrothrix longipilis* Waterhouse, 1837 (17.5%), followed by *Phyllotis darwini* Waterhouse, 1837 (2.7%) and the introduced *Rattus rattus* Linnaeus, 1758 (2.3%). Other small mammals captured were *Irenomys tarsalis* Philippi, 1900, *Octodon bridgesi* Waterhouse 1844 and *T. elegans* (0.7% all three species pooled). 

### 3.2. Parasitological Analysis

In total, 1091 fecal samples were collected and 185 samples (16.95%) were positive to any helminth egg or larvae and were found in the majority of small mammal species analyzed (Table 1 and Table 2). The helminths identified belonged to three phyla: Acanthocephala, Nematoda and Platyhelminthes (Table 1; Figure 1). Another helminth egg was found in several samples from both *Abrothrix* species but could not be identified (Table 1; Figure 1). A larva was also detected in some samples (Table 1) but could not be identified through microscope. The most frequent helminth eggs were *Hymenolepis* sp. (6.1%) and *Physaloptera* sp. (3.5%). Prevalence of helminths in samples of the most abundant rodent species was *A. olivacea* (17.7%), *A. longipilis* (22.2%), *O. longicaudatus* (6.7%) and *P. darwini* (20%) (Table 2). Most helminths were found in both native forest and pine plantations, except Strongylida, which was found only in the native forest, and *Moniliformis* sp., which was only recorded in *Abrothrix* species in both adult and young pine plantations (Table 3).

### 3.3. Data Analysis

GLM analyses were conducted including the results of the three most abundant rodent species (*A. olivacea*, *A. longipilis* and *O. longicaudatus*). *Phyllotis darwini* was not included because this rodent was only present in young pine plantations [15]. The best GLM model selected according to AICc values for all helminths included season and host species as explanatory variables (Appendix A). This model indicates that prevalence of helminths was higher in spring compared to autumn and summer, and *O. longicaudatus* had lower prevalence compared to both *Abrothrix* species (Table 4). Habitat type was included in the third best GLM model (Appendix A), showing that helminth prevalence was not significantly different between native forest and adult pine plantations (estimate = −0.19, standard error (SE) = 0.31, *p* = 0.54) or young pine plantations (estimate = −0.16, SE = 0.30, *p* = 0.59). Regarding GLM analyses using the most common helminth species, the best model for *Hymenolepis* sp. included only host species as an explanatory variable (Appendix A). *Oligoryzomys longicaudatus* had lower prevalence of *Hymenolepis* sp. compared to both *Abrothrix* species (Table 5). Habitat type was included in the second best GLM model (Appendix A), indicating that *Hymenolepis* sp. prevalence was not significantly different between native forest and adult pine plantations (estimate = 0.25, SE = 0.55, *p* = 0.65) or young pine plantations (estimate = 0.46, SE = 0.55, *p* = 0.39). For *Physaloptera* sp., the best model included the variables season and host species (Appendix A), indicating that this parasite had lower prevalence in autumn compared to all other seasons (Table 6). Similar to previous results, habitat type was included in the second best GLM model (Appendix A), showing that *Physaloptera* sp. prevalence was not significantly different between native forest and adult pine plantations (estimate = −1.02, SE = 0.78, *p* = 0.19) or young pine plantations (estimate = 0.52, SE = 0.56, *p* = 0.35). 

## 4. Discussion

In this study, we recorded helminth families or genera that are associated with rodents in Chile and elsewhere [17,35,36]. Furthermore, some of the helminth genera recorded may belong to zoonotic species, such as *Hymenolepis diminuta* Rudolphi 1819, *H. nana* Siebold 1852, *Syphacia obvelata* Rudolphi 1802 and *S. muris* Yamaguti 1941, which have been reported in Chile [30,37]. The following host–parasite associations are registered for the first time: *P. darwini–Physaloptera* sp., *A. longipilis–Moniliformis* sp., *A. longipilis–Physaloptera* sp., *A. longipilis–Capillaria* sp. and *O. longicaudatus–Capillaria* sp. We used traditional routine coprological analyses to detect helminths in feces, which have been successfully employed to address the effect of land use on parasites elsewhere [11,35,38]. However, further studies are needed to identify the helminth species present in the study area which should focus on identification of adult helminths [30] and/or species identification through molecular tools, which has been scarcely used for helminth egg identification from fecal samples of rodents [39].

The findings of some helminths only in native forest (Strongylida in *O. longicaudatus*) or only in pine plantations (*Moniliformis* in *Abrothrix* spp.) might be a consequence of the low prevalence of these parasites (0.37–1.3%, respectively). Our results indicate that helminth prevalence in wild rodents varies among seasons and host species but is not affected by habitat type. Therefore, the replacement of native forest by pine plantations in the study area would not be a driver for increased transmission of helminths with zoonotic potential in rodents. A previous investigation in the study area has shown a similar outcome, in which the prevalence of mites (*Androlaelaps* sp.) parasitizing *A. olivacea* was similar between the native forest and pine plantations [10]. On the other hand, the prevalence of other mite species (*Ornithonyssus* sp.) on the same host was increased in young pine plantations [10]. These contrasting results show that the effect of Monterey pine plantations on parasitism is parasite-dependent, which might be consequence of differences in the ecology, life cycle and other attributes between parasite species and their relation to the environment and their hosts. In fact, the effects of land-use change on helminth parasitism depend on the life history traits of each host–parasite association, as shown in several studies across the world [3,35,40,41].

*Hymenolepis* sp. was the most common helminth found, with higher prevalence in both *Abrothrix* species compared to *O. longicaudatus*. Most species of the genus *Hymenolepis* have an indirect cycle, including free living and ecto-parasite arthropods as intermediate hosts [42]. Therefore, differences in prevalence among rodents could be related to food habits, where the diets of *A. olivacea* and *A. longipilis* include a higher composition of insects (up to 25–32%) [27] compared to *O. longicaudatus,* for which invertebrates represent a minor component of the diet (5−10%) [27,43]. On the other hand, *Physaloptera* sp., the second most prevalent helminth, presented a seasonal variation, with lower prevalence in autumn compared to other seasons. As some *Physaloptera* species have complex cycles with intermediate hosts such as Coleoptera, Blattodea and Orthoptera, seasonal variation could be related to environmental conditions acting on variations in abundance of intermediate hosts and/or dietary changes of hosts, as suggested by Cawthorn and Anderson [44]. In fact, some invertebrates such as ground beetles (Coleoptera: Carabidae) in the study area have seasonal variations, increasing their abundance and richness in summer [45].

Several studies have investigated the impact of Monterrey pine plantation on biodiversity worldwide [25,46]. However, the effect on parasites and pathogens has only been recently addressed [10,15,47]. In this context, more studies are needed for a better understanding on the effects of pine plantations on parasites and pathogens of wildlife, including an ecological approach using multiple temporal and spatial scales [48]. This information will be useful for addressing the nexus between wildlife and zoonosis risks into land use planning, within the One Health framework [49,50]. 

## 5. Conclusions

In this study, we reported several intestinal helminths in fecal samples from wild rodents in the landscape of central Chile, but we did not find differences in helminth prevalence between rodents inhabiting native forests and Monterey pine plantations (adult and young plantations). As Monterey pine plantations are one of the most important forestry plantations worldwide, more research should be conducted in order to better understand the effect of pine plantations on other micro- and macroparasites of public health concern. 

## Figures and Tables

**Figure 1 animals-11-00384-f001:**
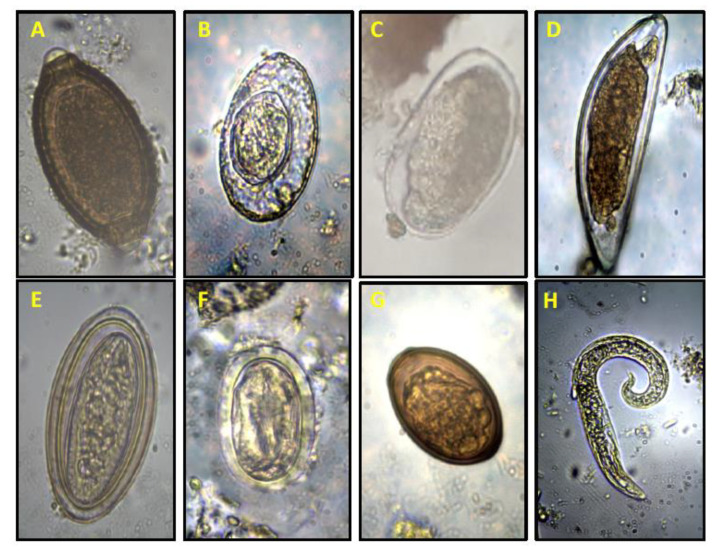
Helminth eggs recorded from fecal samples of rodents. All images were obtained at 40×. (**A**) *Capillaria* sp.; (**B**) *Hymenolepis* sp.; (**C**) Strongylida; (**D**) *Syphacia* sp.; (**E**) *Moniliformis* sp.; (**F**) *Physaloptera* sp.; (**G**) unidentified egg; (**H**) unidentified larva.

**Table 1 animals-11-00384-t001:** Presence of helminth eggs and larvae from fecal samples of small mammals. N = sample size. Number in parentheses is the number of positive samples.

Hosts	N	Acantocephala	Nematoda	Platyhelminthes	Unidentified
*Abrothrix olivacea*	583	*Moniliformis* sp. (11)	*Physaloptera* sp. (30) *Syphacia* sp. (6) *Capillaria* sp. (3) Unidentified larva (8)	*Hymenolepis* sp. (40)	Unidentified egg (2)
*Abrothrix longipilis*	279	*Moniliformis* sp. (3)	*Physaloptera* sp. (3) *Syphacia* sp. (2) *Capillaria* sp. (2) Unidentified larva (4)	*Hymenolepis* sp. (22)	Unidentified egg (20)
*Oligoryzomys longicaudatus*	180	--	*Physaloptera* sp. (3) *Syphacia* sp. (1) *Capillaria* sp. (1) Strongylida (4)	*Hymenolepis* sp. (3)	--
*Phyllotis darwini*	30	--	*Physaloptera* sp. (2) *Syphacia* sp. (2)	*Hymenolepis* sp. (2)	--
*Rattus rattus*	15	--	*Syphacia* sp. (2)	*--*	--
*Thylamys elegans*	3	--	--	--	--
*Octodon bridgesi*	1	--	--	--	--

**Table 2 animals-11-00384-t002:** Helminths in fecal samples of small mammals at three habitat types. Sample size (N), number of positive individuals (+), prevalence (P) and 95% confidence intervals (CI) are reported.

	Native Forest			Adult Pine			Young Pine			Overall		
	N (+)	P (%)	95% CI	N (+)	P (%)	95% CI	N (+)	P (%)	95% CI	N (+)	P (%)	95% CI
*Abrothrix longipilis*	37 (12)	32.4	18–49.8	202 (45)	22.3	16.7–28.6	40 (5)	12.5	4.2–26.8	279 (62)	22.2	17.5–27.6
*Abrothrix olivacea*	31 (2)	6.5	0.8–21.4	48 (8)	16.7	7.5–30.2	504 (93)	18.5	15.2–22.1	583 (103)	17.7	14.7–21.0
*Oligoryzomys longicaudatus*	95 (8)	8.4	3.7–15.9	19 (0)	0	-	66 (4)	6.1	1.7–14.8	180 (12)	6.7	3.5–11.4
*Phyllotis darwini*	0	-	-	0	-	-	30 (6)	20	7.7–38.6	30 (6)	20	7.7–38.6
*Rattus rattus*	7 (1)	14.3	0.4–57.9	5 (1)	20	0.5–71.6	3 (0)	0	-	15 (2)	13	1.7–40.5
*Octodon bridgesi*	1 (0)	0	-	0	-	-	0	-	-	1 (0)	0	-
*Thylamis elegans*	2 (0)	0	-	1 (0)	0	-	0	-	-	3 (0)	0	-
Total	173 (23)	13.3	8.6–19.3	275 (54)	19.7	15.1–24.8	643 (108)	16.8	14–19.9	1091 (185)	16.9	14.8–19.3

**Table 3 animals-11-00384-t003:** Number of infected host individuals reported by helminth species, host species (most abundant species) and by habitat type. NF = native forest; AP = adult pine plantation; YP = Young pine plantation; *n* = number sample size. Numbers in parentheses indicate prevalence.

	*A. longipilis*			*A. olivacea*			*O. longicaudatus*		
	NF *n* = 37	AP *n* = 202	YP *n* = 40	NF *n* = 31	AP *n* = 48	YP *n* = 504	NF *n* = 95	AP *n* = 19	YP *n* = 66
*Hymenolepis* sp.	4 (10.8%)	16 (7.9%)	2 (5%)	0	3 (6.2%)	37 (7.3%)	1 (1.1%)	0	2 (3%)
*Physaloptera* sp.	1 (2.7%)	1 (0.5%)	1 (2.5%)	1 (3.2%)	2 (4.2%)	27 (5.4%)	2 (2.1%)	0	1 (1.5%)
*Syphacia* sp.	1 (2.7%)	1 (0.5%)	0	0	1 (2.1%)	5 (1%)	1 (1.1%)	0	0
*Capillaria* sp.	1 (2.7%)	1 (0.5%)	0	0	1 (2.1%)	12 (2.4%)	0	0	1 (1.5%)
*Moniliformis* sp.	0	2 (1%)	1 (2.5%)	0	1 (2.1%)	10 (2%)	0	0	0
Strongylida	0	0	0	0	0	0	4 (4.2%)	0	0

**Table 4 animals-11-00384-t004:** Results of the best generalized linear model (GLM) that predicted the probability of positivity for helminths in rodents. Spring and *O. longicaudatus* were used as the reference categories. Hosmer–Lemeshow test: χ^2^ = 10.96, *p* = 0.20 (*p* > 0.05 is interpreted as fit).

		Estimate (SE)	*z* Value	*p* Value	Odds Ratio	Lower 95% CI	Upper 95% IC
	Intercept	−2.08 (0.36)	−5.80	<0.001 *			
Season	Autumn	−0.71 (0.24)	−2.98	0.003 *	0.49	0.31	0.78
	Summer	−0.50 (0.23)	−2.18	0.030 *	0.60	0.38	0.95
	Winter	−0.37 (0.25)	−1.50	0.132	0.69	0.43	1.12
Host species	*A. longipilis*	1.21 (0.36)	3.52	<0.001 *	3.36	1.76	6.90
	*A. olivacea*	0.94 (0.33)	2.87	0.004 *	2.57	1.40	5.13

* *p* values < 0.05.

**Table 5 animals-11-00384-t005:** Results of the best GLM that predicted the probability of positivity for *Hymenolepis* sp. in rodents. *O. longicaudatus* was used as the reference category. Hosmer–Lemeshow test: χ^2^ = 5.16, *p* = 0.74 (*p* > 0.05 is interpreted as fit).

		Estimate (SE)	*z* Value	*p* Value	Odds Ratio	Lower 95% CI	Upper 95% CI
	Intercept	−2.01 (0.58)	−7.01	<0.001 *			
Host species	*A. longipilis*	1.62 (0.62)	2.60	0.009 *	5.05	1.72	21.56
	*A. olivacea*	1.47 (0.61)	2.43	0.015 *	4.35	1.56	18.11

* *p* values < 0.05.

**Table 6 animals-11-00384-t006:** Results of the best GLM that predicted the probability of positivity for *Physaloptera* sp. in rodents. Autumn and *O. longicaudatus* were used as the reference categories. Hosmer-Lemeshow test: χ^2^ = 2.43, *p* = 0.97 (*p* > 0.05 is interpreted as fit).

		Estimate (SE)	*z* Value	*p* Value	Odds Ratio	Lower 95% CI	Upper 95% CI
	Intercept	−4.83 (0.71)	−6.80	<0.001 *			
Season	Spring	1.67 (0.60)	2.78	0.005 *	5.31	1.76	19.80
	Summer	1.21 (0.61)	1.98	0.048 *	3.37	1.08	12.75
	Winter	1.37 (0.60)	2.28	0.030 *	3.93	1.30	14.51
Host species	*A. longipilis*	0.90 (0.85)	−1.06	0.290	0.41	0.07	2.33
	*A. olivacea*	0.76 (0.65)	1.18	0.240	2.14	0.68	9.42

* *p* values < 0.05.

## Data Availability

The study did not report any data.

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
