# Peer review of "Intestinal Helminths in Wild Rodents from Native Forest and Exotic Pine Plantations (Pinus radiata) in Central Chile"

_animals, 2021, doi:10.3390/ani11020384_

Round 1

Reviewer 1 Report

The Authors have adequately addressed all the concerns raised by the reviewers to my entire satisfaction. 

The manuscript is worthy of publication in Animals and it can now be accepted in its current form. 

Author Response

Thank you for your comments.

Reviewer 2 Report

The required corrections have been made. I believe that the work can be published.

Author Response

Thank you for your comments.

Reviewer 3 Report

There is no requirement to add an author and date to the genus. However, if this is done please delete the abbreviations “sp.” or/and “spp.”. As a rule, authors are not given by the genre.

Scientific names must not be written arbitrarily. This is regulated i.a. by the International Code of Zoological Nomenclature and Code of Nomenclature for Plants. Please check and change.

Very important: when the original name is changed, e.g. the species is moved to a different genus, both codes use parentheses around the original authority.

57-58 lines: Cryptosporidium spp. Tyzzer, 1907: should be “Cryptosporidium Tyzzer, 1907” or “Cryptosporidium spp.”, the second is obviously better.

57-58 lines: Giardia spp. Kunstler 1882: should be “Giardia Kunstler, 1882” or “Giardia spp.”. The second is obviously better. There is always a comma between the name and the date.

57-58 lines: Schistosoma spp. Weinland, 1858: should be “Schistosoma Weinland, 1858” or “Schistosoma spp.”. Comments see above.

68 line: Monterey pine (Pinus radiata D. Don, 1836) – please check if the author in the original notation was in parentheses (see comment above), if yes should be “Pinus radiata (D. Don, 1836) and with a common name “Monterey pine Pinus radiata (D. Don, 1836)” or “Monterey pine, Pinus radiata (D. Don, 1836)”.

Please check all plants species.

If the first author/authors (with the date) is in parentheses it does not need to be given outside author/authors the parentheses.

73 line: Ornithonyssus sp. Sambon, 1928: should be “Ornithonyssus Sambon, 1928” or “Ornithonyssus sp.”.

E.t.c., e.t.c.

141 line: Llaca mouse-opossum (Thylamys elegans Waterhouse, 1839) – should be Llaca mouse-opossum Thylamys elegans (Waterhouse, 1839). Author in the parentheses.

Here is the page with the correct animal names: https://www.itis.gov/

Please check other animal names.

153-154 lines: Hymenolepis sp. Weinland, 1858 (6.1%) and Physaloptera sp. Rudolphi, 1819 – should be “Hymenolepis Weinland, 1858 and Physaloptera Rudolphi, 1819” or “Hymenolepis sp. and Physaloptera sp.”.

157 line: Strongylida Molin, 1861 - should be” Strongylida”. Please do not give the author and date for families, it is unnecessary.

158 line: Moniliformis sp. Bremser, 1811 -  should  be “Moniliformis Bremser, 1811” or “Moniliformis sp.”.

213-214: A. longipilis-Capillaria sp. Zeder, 1800 – please delete “Zeder, 1800”.

Author Response

This manuscript is a resubmission of an earlier submission. The following is a list of the peer review reports and author responses from that submission.

Round 1

Reviewer 1 Report

Revisions for Animals-896449

The manuscript “Intestinal helminths in wild rodents from native forest and exotic pine plantations (Pinus radiata) in Central Chile” provides data on intestinal parasites in wild rodents from different ecological contexts. The study is interesting and well-written in the English language. However, the Authors should provide more details for parasitological analysis methods and results. Also, the Introduction and Discussion sections could be improved with proper reference to other studies.

The manuscript should be considered for publication in Animals, following major revisions.

Here my commentaries:

Lines 36: Add the word “species” after “small mammals”.

Line 38: Use the plural “associations”.

Line 64: “[but see 9,10]”. Add some details of the main results of these studies. I noticed these were discussed in the “Discussion”, but some information should be also included in the “Introduction” section.

Lines 71-72: The occurrence and the relevance of the main intestinal parasites, not only helminths but also Protozoa, in micromammals, should be more detailed, referring also to the literature outside Chile.

Lines 78-122: For a better understanding, consider dividing the “Materials and methods” section into sub-paragraphs, e.g., Area description (Lines 79-87), Sample collection (Lines 88-101), Parasitological analysis (Lines 108-111), Data analysis (Lines 112-122). The Ethical statement (Lines 102-107) should be moved at the beginning of the section.

Lines 97-98: “animals were identified to species”. Add a proper reference for species identification.

Lines 108-109: A brief description of the coprological method used should be included. Besides, a proper reference of the keys for parasites' identification is needed. A curiosity: did not you detect any Protozoa?  

Lines 123-187: The same division in sub-paragraphs suggested for the “Materials and methods” should be applied also to the “Results” section.

Line 130: “(see more details in Rubio et al. [14]).” These details you referred to with this reference should be provided in the “Discussion”, but I do feel it is not appropriate to cite this study in the “Results” section, where only your own results from your study should be reported.

Lines 134-136: “Another helminth egg was found in several samples from both Abrothrix species, 135 but could not be identified (Table 1; Figure 1). An unidentified larva was also detected in some 136 samples (Table 1).” Did you consider attempting the identification by molecular methods? Figure 1 should also include photos of the larvae.

Lines 188-239: The “Discussion” section could be extended with reference to studies from other countries other than Chile.

Reviewer 2 Report

I think the discussion paragraph needs to be simplified and shorter.

lines 42 - 44: I would completely remove these three lines from "Further ...".
lines 198 - 199: explain why molecular biology methods were not used for a more in-depth investigation of the species identified only morphologically. Explain why a laboratory investigation based on "classical" parasitological medologies is enough.
lines 213 - 219: this speech is not well connected to the rest of the discussion. It needs to be explained better or eliminated.

I think that is a good scientific activity, in particular for the difficulties of finding the materials.
Furthermore, the ability to analyze parasites and eggs found implies a good knowledge and experience in this sector.

Reviewer 3 Report

The study presents the new, interesting data on the helminth species/taxonomic richness of several rodents species from native forest and exotic pine plantations in Chile.

I recommend this paper for publication in the Animal after minor corrections.

1. The aim of this research was not to study the occurrence and prevalence…, but rather to describe the species/taxonomic richness (or diversity) and prevalence... What did the Authors mean by stating “occurrence” ?; the “occurrence” can be associated with prevalence. The word "occurrence" is imprecise here.

2. When a scientific name is used for the first time in the paper (Pinus radiata and other plants, parasites, and host) should use proper nomenclature i.e. author and date when species was published and described.

3. [Introduction, lines 74-77]: I am not sure to use a citation “[23]” in the aim of the study ?

4. [Introduction, lines 72 and Discussion, lines 191-192]:

“… including zoonotic parasites such as Syphacia obvelata, Hymenolepis nana and H. diminuta [21, 22] …” and “zoonotic species, such as Hymenolepis nana, Syphacia obvelata and S. muris, which have been reported in Chile [21, 22]”:

basically the same information but different parasites (Introduction: Syphacia obvelata, Hymenolepis nana and H. diminuta; Discussion: Hymenolepis nana, Syphacia obvelata and S. muris). Please correct.

5. In my opinion, Table S1 should be a permanent element of the paper. The information in this table are closely related to the title and aims of this work.

6. Reference list is inconsistent with regard to Animals (e.g. abbreviated journal name, dashes/hyphens between pages, spaces between colon and pages, ISBN number is not needed, etc.), please see Instructions for Authors.